# Occupational Safety and Health for Adult Saudi Arabian Women: Utilizing National Anthropometric Data

**DOI:** 10.3390/healthcare12010109

**Published:** 2024-01-03

**Authors:** Abdalla Alrashdan

**Affiliations:** Department of Industrial Engineering, College of Engineering, Alfaisal University, Riyadh 11533, Saudi Arabia; aalrshdan@alfaisal.edu

**Keywords:** anthropometric, female population, Saudi Arabia, product and workplace, safety

## Abstract

In the last five years, the female labor force has increased rapidly in Saudi Arabia. This is due to the new government’s vision to empower women. For many decades, Saudi females were excluded from working in certain fields due to cultural restrictions. Nowadays, Saudi women are not only joining the service workforce but are currently employed in more physically demanding careers, such as manufacturing and military jobs, which were previously dominated by males. It becomes necessary to design workplaces, tools, and equipment to safely accommodate the female physical attributes, which include body dimensions. This study presents the anthropometric measurements of Saudi Arabian adult females. In total, 504 female subjects aged 20–70 participated in the study. Thirty-eight body measurements, including weight and triceps skinfold, were taken in sitting and standing postures. The main contribution of this study is to provide a national anthropometric database of Saudi females, which is very limited, especially for females in the age groups under study. The availability of such data will allow foreign and local manufacturers to design usable and safe products and workspaces for a wide range of Saudi adult females. The findings reveal that there are no significant differences in the body dimensions of Saudi females across all age groups, except for stature height, eye height, chest depth, skinfold (mm), sitting height, buttock–knee length, and hip breadth. The study also reveals that Saudi females’ body sizes are different from other Asian, Middle Eastern, and British nations, which invalidates the assumption of using other nations’ body measurements to estimate Saudis’ body measurements. Utilizing the supermarket cashier workstation to assess the appropriateness of commercial station fit for Saudi females’ body dimensions, the results underscore the crucial role of anthropometric measurements in addressing differences between product design and the unique body dimensions of Saudi females. The identified anthropometric mismatch highlights potential risks, emphasizing the threat to the working safety of Saudi females. Moreover, the data can be used by health professionals as a base to evaluate the health of Saudi adult females. Descriptive statistics and extreme values are determined. The data are presented in standard anthropometric tables.

## 1. Introduction

Anthropometry refers to the measurements of the human body dimensions, which can be taken either in static or dynamic states [1,2]. Anthropometric data are essential, as they provide designers with knowledge of the user’s physical dimensions to propose design solutions that fulfill users’ needs [3]. Effective design for high performance and productivity can be achieved by utilizing anthropometric data [4]. The availability of such data is essential for products as well as people-centered spatial design. These data are used to develop adjustment range mechanisms for tools to meet the needs of different users [5,6].

In recent years, anthropometry has been used to design safe and comfortable workplaces and products [7,8]. Employing anthropometry may enhance the well-being of the human body [9]. Health problems such as musculoskeletal disorders are one of the major consequences of mismatching tool dimensions and anthropometric measurements [10,11]. Improper machine and equipment design could increase the chance of work-related injuries and lower work performance [12]. The need for anthropometric data, especially for females, was highly manifested during the COVID-19 pandemic. Women health workers suffered from personal protective equipment (PPE) lack of fit, as this PPE is usually designed for males. According to the Women in Global Health survey [13], more than 85% of female health workers during the pandemic reported that their PPE was not properly designed to fit their bodies, which put them at risk of catching the virus or hindered their work.

Global product design presents challenges for designers, as products should fit humans around the globe under different environmental operations. Various studies have investigated the impact of body measurements on the design of products/workstations. For example, Wang et al. [14] collected anthropometry data from Taiwanese workers and developed standardized dimensions for workplace layout planning. Hanson et al. [15] presented the utilization of anthropometry for a workplace design application in Sweden.

The recent literature has extensively explored women’s anthropometry in the context of minimizing adverse occupational health outcomes. Studies such as [16] have focused on addressing the mismatch of workplace height for manual handling, utilizing female anthropometric measures to reduce musculoskeletal disorders. The design of manual tools for various occupations, including farming [17], hand looming [18], dentistry [19], and packaging [20], has integrated women’s anthropometry to mitigate fatigue and biomechanical stresses. Furthermore, the application of anthropometric measurements extends beyond civilian occupations to active duty, where studies [21,22,23,24] have utilized female anthropometric data to design ergonomic body armor tailored to their body measurements, enhancing battlefield operational performance. Additionally, investigations into the work routines of females, including the impact of rotating shifts on body dimensions [25], contribute to a comprehensive understanding of the intersection between anthropometry and women’s occupational health.

New trends of research consider human anthropometry in product design and evaluation of the product design lifecycle, such as in the areas of protective equipment and school furniture [26,27]. With the advancement of the industry 4.0 work environment, researchers have started to explore anthropometric data for different applications, such as touch screen information systems design [28], potential employee selection [29], and ergonomic product determination [30].

Anthropometric data are expected to be different among races and nationalities [31]. For that, anthropometric data should be available for different populations varying in gender, age, and ethnicity. Acquiring anthropometrics for a specific population is very challenging, as it is usually collected voluntarily, and it is hard to find enough subjects who are willing to participate. There have been many attempts to report national anthropometric data by different researchers from the United States [32], Britain [33], Turkey [34], Taiwan [35], Poland [36], Singapore and Indonesia [37,38], Sweden [15], Uganda [39], Iran [10], Malaysia [40], and many more. 

There have been few attempts to develop international databases to provide anthropometric measurements for product design, such as the Civilian American and European Surface Anthropometry Resource Project (CAESAR) and the World Engineering Anthropometry Resource (WEAR). While CEASAR charges high fees to access the data, WEAR requires a subscription. Moreover, the International Organization for Standardization (ISO7250) published a technical report that includes basic human body measurements for technological design from different countries worldwide [41]. These databases are limited and need continuous updates to describe the body dimensions of the future generation. To decrease the efforts of collecting a comprehensive anthropometric database, new techniques are developed to create a database for a particular population based only on a few available body measurements, presented as percentiles [42].

For many years, Saudi females’ workforce participation was very minimal, especially in the manufacturing sector, which was mainly dominated by their male counterparts. In recent years, the participation of women in the workforce has increased rapidly due to government policies to empower women. As of the third quarter of 2022, Saudi females comprised 37 percent of the total workforce according to the General Authority for Statistics [43]. This figure has more than doubled in the last five years.

Saudi Arabia is like other countries in the region, with few anthropometric studies. Examples of anthropometric studies in the region include research from Saudi Arabia [44,45,46,47], Qatar [48], and Bahrain [49]. The objective of this research is to present anthropometric data on women in Saudi Arabia. The sample covers a wide range of women aged 20–70 years. The data is expected to be utilized by designers and engineers to design safe and efficient products and equipment suitable for Saudi females. Moreover, triceps skinfold measurement is surveyed to mark the health of Saudi females. 

## 2. Materials and Methods

### 2.1. Participants

The participants were recruited from Riyadh, the capital of Saudi Arabia. It is the highest populated city and the main financial and government hub of the country. The city is assumed to represent the country’s demographics, as it is a mixture of Saudis who moved from different regions to Riyadh seeking job opportunities. It was difficult to recruit participants voluntarily, as women were very hesitant to reveal their body sizes. The participants were solicited by visiting different work facilities, such as hospitals, banks, grocery stores, universities, restaurants, households, etc. The randomness of the sample was sought as much as possible. However, some subjects, especially older participants, were selected to complete the data set of all possible categories under study. Inclusion criteria included participants who have not experienced any neuromuscular disease or injury and had no recent or ongoing hand or upper-limb injury nor any normal mobility problems. Before the study, ethical clearance was obtained from the Institutional Review Board at Alfaisal University.

The data were collected from women between 20 and 70 years old. The sample is divided into 4 different age groups (20–29, 30–39, 40–49, and 50–70). A total of 504 subjects participated in the study, with 126 subjects for each age group range being selected for measurement.

### 2.2. Sample Size

The determination of the sample size follows the procedures outlined in the International Organization for Standardization (ISO) 15535—General Requirements for Establishing Anthropometric Databases [50], as illustrated in Equations (1) and (2).
(1)$$N={\left(\frac{1.96\times CV}{\alpha }\right)}^{2}\times 1.534^{2}$$
(2)$$CV=\frac{SD}{\bar{x}}\times 100$$
where *N* represents the number of individuals in the sample, *CV* is the coefficient of variance, and *α* denotes the desired percentage of relative accuracy. Given the fixed number of individuals (*N* = 126) in this study due to sampling limitations, the percentage of relative accuracy can be computed using Equation (3).
(3)$$\alpha =\frac{1.96\times CV}{\sqrt{\frac{126}{1.534^{2}}}}$$

According to [51], the percentage of relative accuracy is selected as between 2 to 5 percent. In our study, *α* was found in this range, except for skinfold measurement, where it ranged between 6 and 8 percent.

### 2.3. Body Dimension

The body measurements in standing and sitting postures were collected as shown in Figure 1. Thirty-eight common anthropometric body dimensions were selected based on their relevance to engineering designs and cultural acceptance from a measurement point of view. Measurements and terminologies in both standing and sitting postures were utilized, as explained in ISO7250 [41]. Moreover, triceps skinfold (TSF) measurements were surveyed, as shown in Figure 2.

### 2.4. Equipment

It is worth noting that there are a variety of methods for measuring body dimensions for equipment design. Some of these approaches, such as three-dimensional scanners, are costly, complex, and not accessible to all researchers; others, such as traditional anthropometric equipment, are simple and inexpensive. Furthermore, traditional measures yielded anthropometric data that showed no difference from that obtained through 3D scanners [52]. In this study, A RossCraft Anthropometric kit, which includes calipers, a segmometer, tapes, a stadiometer, and a skinfold caliper, was used to measure body dimensions as shown in Figure 3.

### 2.5. Procedure

Two female experimenters were trained to become accustomed to the measurement tools and procedures according to the standards described in ISO7250 [41]. The female experimenters are students who have successfully completed a course in ergonomics, which covered three laboratory experiments focused on collecting anthropometric measurements within a class of 130 students. Additionally, they underwent a supplementary five-day training program to enhance their proficiency in conducting measurements on subjects and to evaluate and ensure the inter-measurer reliability of their assessments.

A comprehensive reliability study on all anthropometric measurements for 20 subjects was conducted. Each participant underwent two sets of measurements, ensuring a minimum of 25 degrees of freedom in accordance with the guidelines established in [53]. The focus of the investigation involved both intra-rater and inter-rater reliability, evaluating the consistency within each experimenter’s measurements and the agreement between different experimenters, respectively. The minimum Intraclass Correlation Coefficient (ICC) for intra-rater reliability was found to be 0.88. Simultaneously, the inter-rater reliability study also yielded a minimum ICC of 0.82.

The precision and applicability of anthropometric instrument measurements were ensured by following the guidelines outlined in reference [54] for calipers. These recommendations contain the proper assembly of the device, careful dimension taking, and accurate recording of measurements. Standardization protocols were implemented to address concerns related to misalignment and the applied force during measurements. For skinfold caliper measurements and device calibrations, accuracy was achieved by adhering to the guidelines set forth by the RossCraft recommendations [55].

The subjects were fully informed of the measurement procedure and the purpose of the study. The experimenters located the body landmarks and used proper tapes or calipers to take the measurements. Measurements were always taken on the right-hand side of the individuals to achieve more scientific uniformity. Except for the measures of skinfold thickness, which were taken three times (average is recorded), only one measurement was recorded per body dimension. Standing posture measurements were collected while the subject was standing straight against a wall, as shown in Figure 4. For the sitting posture measurements, each subject was sitting in an upright position on an adjustable horizontal surface seat with their knees bent at 90 degrees and their feet touching the floor. Measurements were taken to the nearest millimeter and were recorded in centimeters, except for the skin fold, which was measured and recorded in millimeters. Subjects were barefooted and wearing light clothing during the measurement process. Normally, the investigators worked in a private room to provide the subject with the most preferable environment. All subjects were provided with a non-disclosure agreement to safeguard their names. Thus, all participants signed an informed consent form.

### 2.6. Data Analysis

Statistical analysis of the data was performed with SPSS software version 23 (SPSS Inc., Chicago, IL, USA). In all analyses, *p*-values < 0.05 were considered statistically significant. Tukey’s test was used to determine the significant effect of age group on the body dimensions. The normality of data was tested using the Shapiro–Wilk test.

## 3. Results

In total, measurements of 38 anthropometric dimensions, including weight and triceps skinfold, were recorded for 504 Saudi females in both sitting and standing postures. Descriptive statistics, including the mean and standard deviation, are provided for each age group in both standing and sitting positions. These statistical summaries are presented in Table 1, Table 2, Table 3 and Table 4, which also include the standard error of the mean (SEM) and the coefficient of variation (CV). Furthermore, to evaluate the health indicators of Saudi females, Relative Sitting Height (RSH = sitting height/stature), Body Mass Index (BMI = body weight (kg) divided by squared stature (m)), and Body Surface Area (BSA = 0.007184 × (Height(cm)^0.725) × (Weight(kg)^0.425)) are computed using the gathered data. The outcomes for each group are presented in Table 5.

## 4. Discussion

This study provides anthropometric data for the Saudi female population. The presented percentile values serve as valuable references for guiding safe product and various workplace designs, particularly for Saudi females, who have recently become extensively engaged in the workforce. Depending on anthropometric data from other populations may result in mismatches that could potentially lead to adverse health effects. It is imperative to consider factors such as gender, age, ethnicity, and occupation to align the designs of products, environments, and systems holistically.

### 4.1. Comparison of Anthropometric Data for Saudi Females by Age Group

The Tukey test was performed to determine the differences between age groups for all body dimensions. We only presented the analysis results of body dimensions that were significant. Results showed that age groups have a significant effect on the stature height, eye height, chest depth, skinfold (mm), sitting height, buttock–knee length, and hip breadth of the participants. Table 6 illustrates the effects of age groups on stature height, eye height, chest depth, skinfold (mm), sitting height, buttock–knee length, and hip breadth. In the post hoc analysis, the age groups were grouped to show the significant differences among one another, as shown in Table 7.

### 4.2. Comparison of Average Anthropometric Dimensions in Saudi Arabia with Asian Countries

To assess the validity of utilizing anthropometric data from diverse populations as a representation for the Saudi population, a comparative analysis is conducted. This aims to examine the assumption that measurements from various Asian (e.g., Singaporean [6], Indonesian [38], Thai [4], Taiwanese, Chinese, Japanese, and Korean [56]), Middle Eastern (including Iranians [10], Egyptians [57], and Omanis [58]), and European (specifically British [33]) populations might be suitable for estimating Saudi anthropometric dimensions. The results of this comparison are presented in Table 8, providing insights into potential variations in anthropometric profiles among these populations.

In the comparison of weight, Middle Eastern females exhibit higher weight than other populations, with Omani females ranking highest and Saudi females following closely. In contrast, Thai females have the lowest weight among the populations considered. Analyzing body dimensions in a standing posture reveals that Indonesians have the tallest stature and elbow height, while the Japanese and Chinese share the smallest stature, elbow height, eye height, and shoulder height. Singaporeans present the highest eye height and shoulder height. Saudi and Taiwanese females boast the largest knuckle height and chest depth, while Egyptians have the smallest knuckle height, and Thai females exhibit the smallest chest depth.

In the sitting posture, Saudi and Omani females have the lowest sitting height, while Omanis have the smallest sitting eye height, and Egyptians exhibit the smallest elbow rest height. Korean females possess the greatest sitting height, sitting eye height, and elbow rest height. Omani females lead in knee height, and Saudi females have the greatest popliteal height and elbow-to-elbow breadth. Iranians boast the greatest thigh clearance, the British females have the greatest buttock–knee length, and Omanis top the list for the greatest hip breadth. Ethnic disparities in body shape stem from hereditary effects, economic development, social environment, job type, and labor structure.

### 4.3. Product Mismatch Highlight

To highlight the importance of measuring female dimensions, we investigated the mismatch between workstation design and the body sizes of Saudi females. We investigated the cashier workstations in supermarkets as an example. The cashier career is experiencing increased interest among Saudi females, a shift that was not culturally accepted in the past. Today, many Saudi females are actively working as cashiers in supermarkets. However, the existing mismatch in workstation design presents significant challenges. The combination of high levels of material handling and sustained awkward postures over extended periods poses a heightened risk of developing cumulative trauma disorders [59].

The typical workstation in a supermarket is illustrated in Figure 5. It includes a receiving area with a powered belt, scanner, cash register, and bagging area. Our investigation into the mismatch focuses on the height of the receiving area. The mismatch analysis is based on the method of limits explained in [60]. The working area’s height is segmented into zones, covering unacceptable, acceptable, and optimal ranges. This segmentation is defined by the worker’s elbow height (EH) and the optimum working height criteria presented in [61]. As stated in reference [60], the optimal working height is recommended to be positioned between 50 to 100 mm below the elbow height (EH). Additionally, two other criteria, EH-50 and EH-150, are considered acceptable [60]. These specifications are illustrated in Figure 6.

The elbow height for Saudi females is detailed in Table 1, Table 2, Table 3 and Table 4. Assuming a normal distribution for elbow height, the four regions outlined by the four criteria are also presumed to follow a normal distribution. Using elbow height data from Saudi females across all age groups and considering the insignificance of differences in elbow height between age groups, the mean and standard deviation of Saudi female elbow height while standing are calculated to be 1004 mm and 61 mm, respectively. Accounting for a 25 mm shoe height, the normal distribution for the four criteria is established as shown in Table 9. To assess the mismatch, we examine a commonly used exported commercial cashier station found in various supermarkets, with a height of the receiving area of approximately 900 mm. The results, outlined in Table 9, reveal the percentage distribution of the Saudi female population across each region, considering the receiving area’s height as set at 900 mm. The calculations indicate that 37% of female cashiers will find the height much too low, 68% will find it too low, 10% will find it too high, and 2% will find it much too high, while only 28% will find it comfortable. It is worth noting that a mere increase in height of only 50 mm will result in a 10% rise in the percentage of Saudi females within the comfort zone. According to [59], the optimal working area height should be the average elbow height plus 75 mm, which, in the case of Saudi females, is approximately 1080 mm.

It is evident from the preceding discussion that there is a clear necessity for anthropometric measurements to prevent any mismatch between product design, work layout, and body dimensions, thereby avoiding adaptations that could lead to musculoskeletal disorders. Ignoring these anthropometric considerations poses a significant risk and jeopardizes the working safety of Saudi females.

## 5. Conclusions

Anthropometric data of Saudi females aged 20–70 years were surveyed and presented in standard anthropometric tables. Thirty-eight different body size variables, including skinfold, were measured. The studied age domain was divided into four age groups. Statistical analyses were utilized to determine the changes in the body sizes across the age groups. It was found that the age groups have significant differences in Stature Height, Eye Height, Chest Depth, Skin Fold, Sitting Height, Buttock–Knee Length, and Hip Breadth, and the anthropometric tables were constructed based on the data collected from age 20 to age 70. Moreover, The Saudi females’ body dimensions were compared with similar populations from other Asian nationalities. It was found that Saudi females are different in many body measurements than their Asian female counterparts, which supports the necessity of developing an anthropometric database for different Saudi populations. The study is believed to be the first anthropometric study in the age domain of 20–70 for the female population in Saudi Arabia.

These results are expected to fill a gap in the anthropometric research in the country. This study is expected to be utilized in the design process to design products or systems that safely fit Saudi working females. Moreover, the skinfold study is expected to establish a base for healthcare professionals for evaluating Saudi females’ health. The study faced challenging problems due to cultural restrictions in recruiting females to participate voluntarily in the study. This study is expected to drive researchers at the national level to continue working in the area to include other groups of the Saudi population and to investigate larger sample sizes.

## 6. Limitations

A wider anthropometric data collection will be necessary to increase the representation of the entire Saudi Arabia population in the future. Another limitation of our study is that it was only undertaken in one city, the capital, although the capital usually contains residents representing all parts of the country due to the availability of services and job opportunities.

## Figures and Tables

**Figure 1 healthcare-12-00109-f001:**
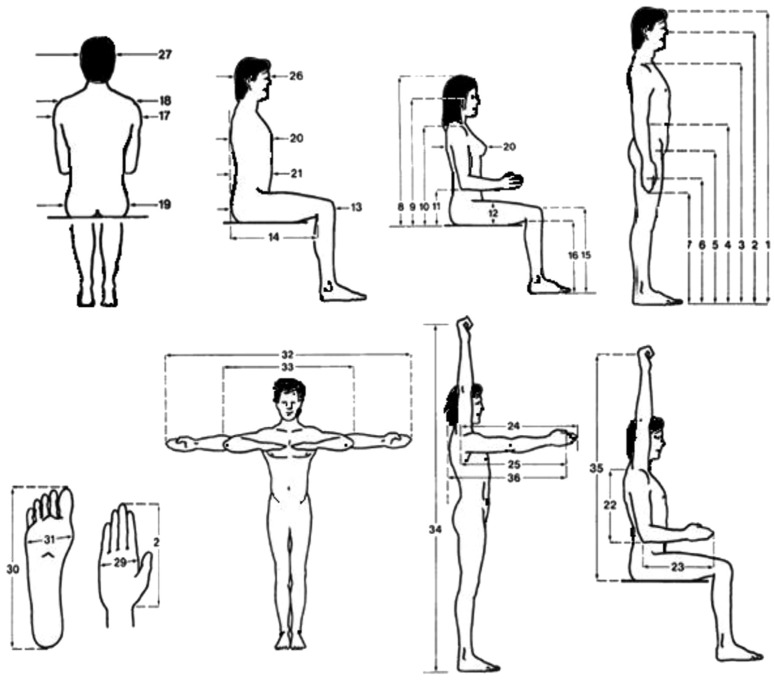
Body measurement in standing and sitting; courtesy of [10].

**Figure 2 healthcare-12-00109-f002:**
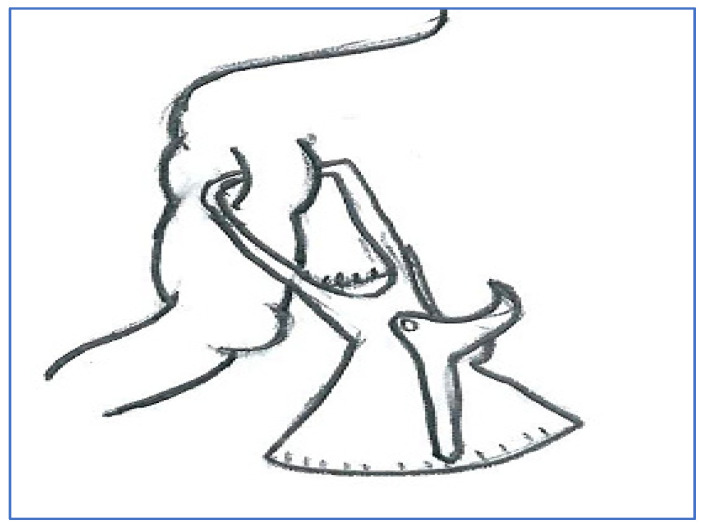
Triceps skinfold measurement.

**Figure 3 healthcare-12-00109-f003:**
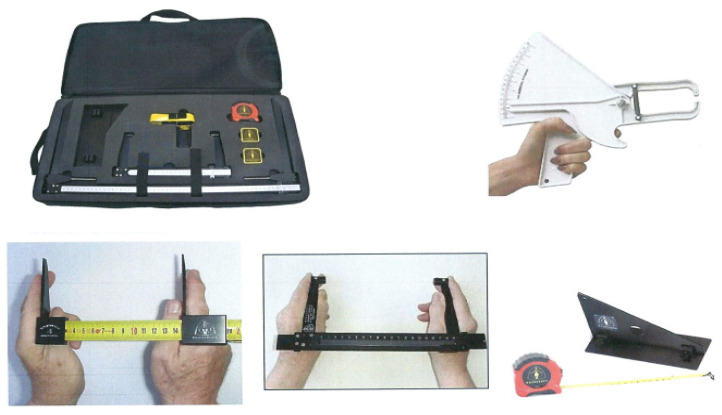
Anthropometry equipment used for body measurements.

**Figure 4 healthcare-12-00109-f004:**
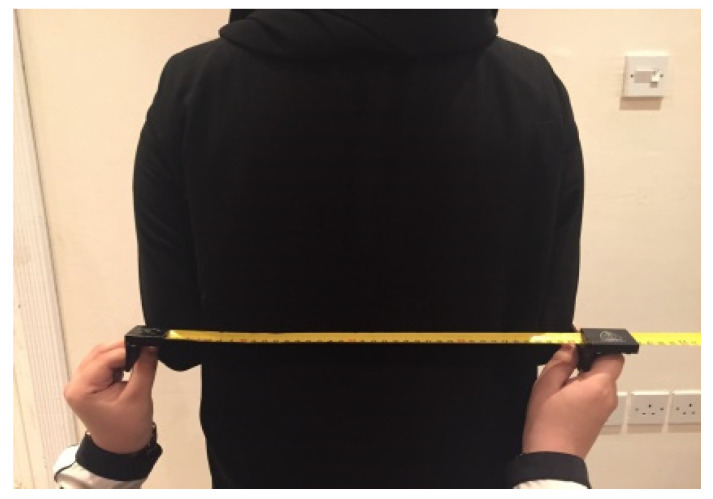
Technicians measuring participants’ body dimensions.

**Figure 5 healthcare-12-00109-f005:**
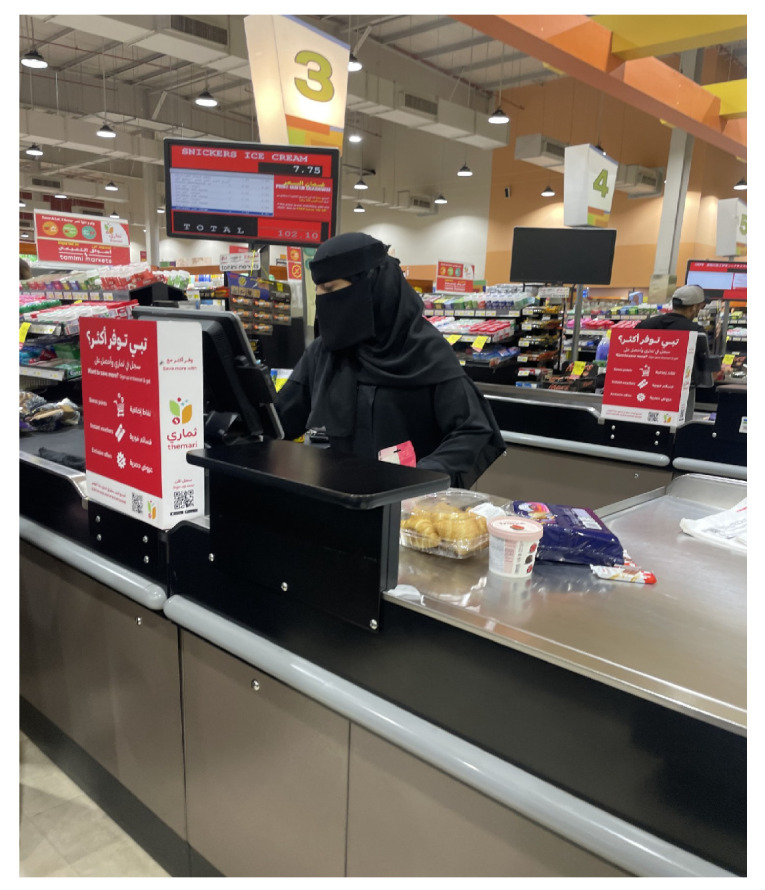
A typical supermarket workstation layout.

**Figure 6 healthcare-12-00109-f006:**
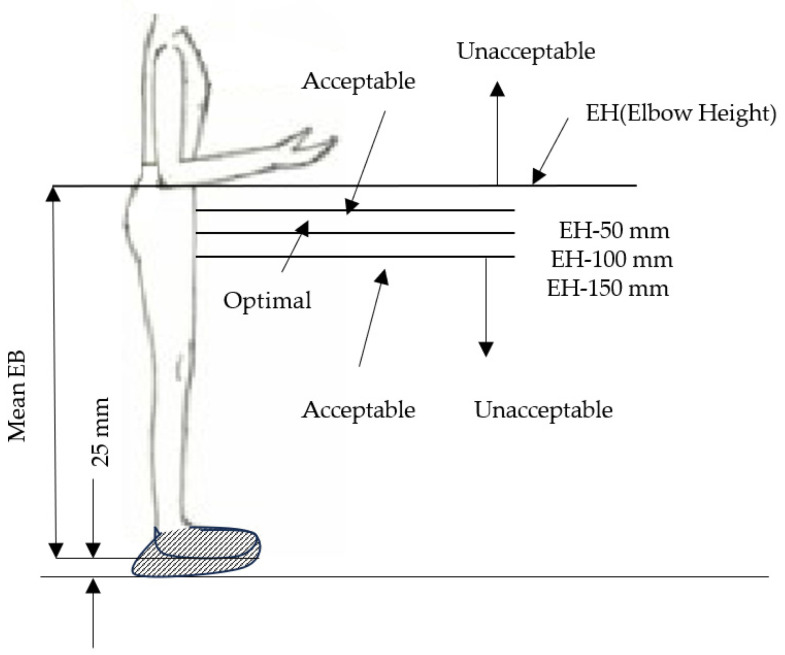
Criteria for optimal and acceptable working heights.

**Table 1 healthcare-12-00109-t001:** Anthropometric data for Saudi Arabia females, aged 20–29 years (n = 126).

Dimensions	Mean	SD	SEM	5th Percentile	95th Percentile	CV
1. Stature Height	160.72	7.33	0.65	148.70	172.73	0.05
2. Eye Height	150.94	7.43	0.66	138.76	163.12	0.05
3. Shoulder Height	134.03	6.34	0.56	123.64	144.42	0.05
4. Elbow Height	103.07	6.50	0.58	92.42	113.73	0.06
5. Hip Height	86.31	3.71	0.33	80.23	92.39	0.04
6. Knuckle Height	71.36	3.86	0.34	65.95	80.70	0.05
7. Fingertip height	59.64	2.37	0.21	55.7532	63.5268	0.04
8. Sitting Height	82.12	12.54	1.12	61.55	102.68	0.15
9. Sitting Eye Height	71.35	4.80	0.43	63.47	79.22	0.07
10. Sitting Shoulder Height	59.94	3.96	0.35	53.46	66.43	0.07
11. Elbow Rest Height	22.87	5.58	0.50	13.71	32.03	0.24
12. Thigh Clearance	13.12	3.63	0.32	7.17	19.07	0.28
13. Buttock–Knee Length	53.16	5.56	0.50	44.05	62.28	0.10
14. Buttock–Popliteal Length	43.21	5.52	0.49	34.16	52.26	0.13
15. Knee Height	47.75	3.93	0.35	41.30	54.20	0.08
16. Popliteal Height	36.57	3.25	0.29	31.24	41.90	0.09
17. Shoulder Breadth (Bideltoid)	39.36	1.71	0.15	36.56	42.16	0.04
18. Shoulder Breadth (Biacromial)	30.44	1.89	0.17	27.34	33.55	0.06
19. Hip Breadth	35.62	5.68	0.51	26.30	44.94	0.16
20. Chest Depth	19.21	4.91	0.44	11.16	27.26	0.26
21. Abdominal Depth	19.00	2.47	0.22	14.94	23.06	0.13
22. Shoulder–Elbow Length	33.61	1.50	0.13	31.16	36.06	0.04
23. Elbow–Fingertip Length	41.22	2.03	0.18	37.89	44.56	0.05
24. Upper Limb Length	70.72	2.71	0.24	66.29	75.16	0.04
25. Shoulder–Grip Length	58.39	5.01	0.45	50.17	66.61	0.09
26. Head Length	17.26	0.36	0.03	16.66	17.85	0.02
27. Head Breadth	13.59	0.52	0.05	12.74	14.43	0.04
28. Hand Length	16.83	0.94	0.08	15.30	18.37	0.06
29. Hand Breadth	7.89	0.22	0.02	7.53	8.25	0.03
30. Foot Length	23.90	2.63	0.23	19.59	28.21	0.11
31. Foot Breadth	9.12	1.62	0.14	6.46	11.78	0.18
32. Span	155.89	7.93	0.71	142.89	168.89	0.05
33. Elbow Span	85.31	5.28	0.47	76.65	93.97	0.06
34. Vertical Grip Reach (Standing)	195.15	8.56	0.76	181.11	209.19	0.04
35. Vertical Grip Reach (Sitting)	116.1	4.65	0.41	108.47	123.73	0.04
36. Forward Grip Reach	67.11	4.57	0.41	59.62	74.60	0.07
37. Skin Fold (mm)	26.70	9.43	0.84	11.23	42.16	0.35
38. Weight	69.85	14.32	1.28	46.36	93.33	0.21

**Table 2 healthcare-12-00109-t002:** Anthropometric data for Saudi Arabia females, aged 30–39 years (n = 126).

Dimensions	Mean	SD	SEM	5th Percentile	95th Percentile	CV
1. Stature Height	158.40	6.69	0.60	147.42	169.38	0.04
2. Eye Height	147.98	11.36	1.01	129.35	166.61	0.08
3. Shoulder Height	133.06	6.43	0.57	122.52	143.61	0.05
4. Elbow Height	99.62	14.19	1.26	76.34	122.89	0.14
5. Hip Height	81.33	4.14	0.37	74.54	88.12	0.05
6. Knuckle Height	69.76	3.37	0.30	64.23	75.29	0.05
7. Fingertip height	57.83	2.28	0.20	54.09	61.57	0.04
8. Sitting Height	79.91	4.16	0.37	73.09	86.73	0.05
9. Sitting Eye Height	70.06	6.82	0.61	58.87	81.25	0.10
10. Sitting Shoulder Height	58.79	1.95	0.17	55.58	61.99	0.03
11. Elbow Rest Height	22.29	3.21	0.29	17.02	27.56	0.14
12. Thigh Clearance	13.22	4.89	0.44	5.19	21.24	0.37
13. Buttock–Knee Length	54.35	5.28	0.47	45.68	63.01	0.10
14. Buttock–Popliteal Length	44.32	4.75	0.42	36.53	52.11	0.11
15. Knee Height	48.68	4.39	0.39	41.48	55.88	0.09
16. Popliteal Height	37.96	2.94	0.26	33.14	42.78	0.08
17. Shoulder Breadth (Bideltoid)	37.56	1.84	0.16	34.54	40.58	0.05
18. Shoulder Breadth (Biacromial)	29.71	1.91	0.17	26.58	32.85	0.06
19. Hip Breadth	36.75	4.40	0.39	29.53	43.97	0.12
20. Chest Depth	21.37	5.07	0.45	13.05	29.68	0.24
21. Abdominal Depth	20.86	1.93	0.17	17.69	24.02	0.09
22. Shoulder–Elbow Length	34.43	3.51	0.31	28.68	40.18	0.10
23. Elbow–Fingertip Length	41.50	1.41	0.13	39.18	43.82	0.03
24. Upper Limb Length	68.93	1.02	0.09	67.26	70.60	0.01
25. Shoulder–Grip Length	54.29	2.23	0.20	50.62	57.95	0.04
26. Head Length	17.00	0.65	0.06	15.94	18.06	0.04
27. Head Breadth	13.60	0.67	0.06	12.50	14.70	0.05
28. Hand Length	17.36	0.63	0.06	16.33	18.39	0.04
29. Hand Breadth	7.71	0.27	0.02	7.28	8.15	0.04
30. Foot Length	23.69	0.95	0.08	22.13	25.25	0.04
31. Foot Breadth	9.44	1.34	0.12	7.24	11.64	0.14
32. Span	155.64	6.46	0.58	145.05	166.24	0.04
33. Elbow Span	86.82	4.68	0.42	79.14	94.50	0.05
34. Vertical Grip Reach (Standing)	187.37	1.04	0.27	185.66	189.08	0.02
35. Vertical Grip Reach (Sitting)	113.02	2.15	0.19	121.17	128.60	0.02
36. Forward Grip Reach	65.54	7.89	0.70	52.61	78.47	0.12
37. Skin Fold (mm)	32.40	10.34	0.92	15.45	49.35	0.32
38. Weight	61.30	9.72	0.87	45.36	77.24	0.16

**Table 3 healthcare-12-00109-t003:** Anthropometric data for Saudi Arabia females, aged 40–49 years (n = 126).

Dimensions	Mean	SD	SEM	5th Percentile	95th Percentile	CV
1. Stature Height	157.38	6.49	0.58	146.74	168.02	0.04
2. Eye Height	147.83	6.68	0.60	136.88	158.78	0.05
3. Shoulder Height	132.73	5.99	0.53	122.91	142.56	0.05
4. Elbow Height	103.09	9.90	0.88	86.86	119.33	0.10
5. Hip Height	78.28	2.62	0.23	73.98	82.58	0.03
6. Knuckle Height	70.75	2.96	0.26	65.90	75.60	0.04
7. Fingertip height	57.82	3.13	0.28	52.69	62.95	0.05
8. Sitting Height	78.79	3.94	0.35	72.32	85.25	0.05
9. Sitting Eye Height	69.97	5.27	0.47	61.33	78.61	0.08
10. Sitting Shoulder Height	47.83	4.65	0.41	40.21	55.45	0.10
11. Elbow Rest Height	23.11	4.21	0.38	16.20	30.02	0.18
12. Thigh Clearance	13.33	3.23	0.29	8.02	18.63	0.24
13. Buttock–Knee Length	53.70	4.21	0.38	46.80	60.60	0.08
14. Buttock–Popliteal Length	43.67	4.15	0.37	36.86	50.48	0.10
15. Knee Height	46.3	3.31	0.29	40.87	51.73	0.07
16. Popliteal Height	38.17	3.35	0.30	32.68	43.66	0.09
17. Shoulder Breadth (Bideltoid)	43.67	4.15	0.37	36.86	50.48	0.10
18. Shoulder Breadth (Biacromial)	30.67	4.16	0.37	23.84	37.49	0.14
19. Hip Breadth	38.17	4.98	0.44	30.00	46.33	0.13
20. Chest Depth	20.41	4.72	0.42	12.67	28.15	0.23
21. Abdominal Depth	21.67	4.04	0.36	15.04	28.29	0.19
22. Shoulder–Elbow Length	32.50	2.50	0.22	28.40	36.60	0.08
23. Elbow–Fingertip Length	39.67	2.52	0.22	35.54	43.79	0.06
24. Upper Limb Length	70.50	3.91	0.35	64.10	76.90	0.06
25. Shoulder–Grip Length	61.33	4.51	0.40	53.94	68.73	0.07
26. Head Length	16.67	0.58	0.05	15.72	17.61	0.03
27. Head Breadth	13.37	0.32	0.03	12.84	13.89	0.02
28. Hand Length	17.83	0.76	0.07	16.58	19.09	0.04
29. Hand Breadth	7.83	0.29	0.03	7.36	8.31	0.04
30. Foot Length	24.13	1.11	0.10	22.31	25.95	0.05
31. Foot Breadth	10.23	0.75	0.07	9.00	11.46	0.07
32. Span	161.33	6.11	0.54	151.31	171.35	0.04
33. Elbow Span	84.49	4.13	0.37	77.72	91.26	0.05
34. Vertical Grip Reach (Standing)	192.53	3.35	0.30	187.04	198.03	0.02
35. Vertical Grip Reach (Sitting)	114.25	2.012	0.18	110.95	117.55	0.02
36. Forward Grip Reach	66.32	4.45	0.40	63.48	78.52	0.07
37. Skin Fold (mm)	34.78	11.53	1.03	15.87	53.69	0.33
38. Weight	73.20	11.70	1.04	54.01	92.39	0.16

**Table 4 healthcare-12-00109-t004:** Anthropometric data for Saudi Arabia females, aged 50–70 years (n = 126).

Dimensions	Mean	SD	SEM	5th Percentile	95th Percentile	CV
1. Stature Height	159.03	7.81	0.70	146.22	171.84	0.05
2. Eye Height	155.51	6.04	0.54	145.60	165.42	0.04
3. Shoulder Height	131.62	5.62	0.50	122.40	140.84	0.04
4. Elbow Height	96.8	6.07	0.54	86.85	106.75	0.06
5. Hip eight	84.54	4.1	0.37	77.82	91.26	0.05
6. Knuckle Height	70.57	3.04	0.27	65.58	75.56	0.04
7. Fingertip height	57.5	2.7	0.24	53.07	61.93	0.05
8. Sitting Height	78.62	5.52	0.49	69.57	87.67	0.07
9. Sitting Eye Height	69.89	5.04	0.45	61.62	78.16	0.07
10. Sitting Shoulder Height	55.5	0.71	0.06	54.34	56.66	0.01
11. Elbow Rest Height	23.34	2.55	0.23	19.16	27.52	0.11
12. Thigh Clearance	13.07	2.9	0.26	8.31	17.83	0.22
13. Buttock–Knee Length	50.29	4.58	0.41	42.78	57.80	0.09
14. Buttock–Popliteal Length	43.2	4.56	0.41	35.72	50.68	0.11
15. Knee Height	46.66	4.97	0.44	38.51	54.81	0.11
16. Popliteal Height	38.55	2.57	0.23	34.34	42.76	0.07
17. Shoulder Breadth (Bideltoid)	36.89	4.01	0.36	30.31	43.47	0.11
18. Shoulder Breadth (Biacromial)	28	1.41	0.13	25.69	30.31	0.05
19. Hip Breadth	36.35	4.16	0.37	29.53	43.17	0.11
20. Chest Depth	22.65	4.7	0.42	14.94	30.36	0.21
21. Abdominal Depth	28.5	1.98	0.18	25.25	31.75	0.07
22. Shoulder–Elbow Length	32.5	1.48	0.13	30.07	34.93	0.05
23. Elbow–Fingertip Length	38.5	3.1	0.28	33.42	43.58	0.08
24. Upper Limb Length	68.5	4.1	0.37	61.78	75.22	0.06
25. Shoulder–Grip Length	56.57	5.1	0.45	48.21	64.93	0.09
26. Head Length	18	1.41	0.13	15.69	20.31	0.08
27. Head Breadth	13.5	2.28	0.20	9.76	17.24	0.17
28. Hand Length	15	1.41	0.13	12.69	17.31	0.09
29. Hand Breadth	7.5	0.58	0.05	6.55	8.45	0.08
30. Foot Length	23.26	1.41	0.13	20.95	25.57	0.06
31. Foot Breadth	9.33	1.24	0.11	7.30	11.36	0.13
32. Span	158.1	6.52	0.58	147.41	168.79	0.04
33. Elbow Span	80.8	6.54	0.58	70.07	91.53	0.08
34. Vertical Grip Reach (Standing)	189.89	3.77	0.34	183.71	196.07	0.02
35. Vertical Grip Reach (Sitting)	111.54	0.96	0.09	109.97	113.11	0.01
36. Forward Grip Reach	63.04	5.56	0.50	53.92	72.16	0.09
37. Skin Fold (mm)	25.54	5.65	0.50	16.27	34.81	0.22
38. Weight	75.1	3.82	0.34	68.84	81.36	0.05

**Table 5 healthcare-12-00109-t005:** BMI, RSH, and BSA for Saudi Arabia female age groups.

Age Group	BMI (Kg/m^2^)	RSH	BSA (m^2^)
20–29	27.04	0.51	1.74
30–39	24.43	0.50	1.63
40–49	29.55	0.50	1.74
50–70	29.69	0.49	1.78

**Table 6 healthcare-12-00109-t006:** One-way analysis of variance of age group measurements.

Dimensions	Squares Sum Mean	DF Source, Error	Mean Square	F	*p*
Stature height	881.44	3297	293.81	6.467	0.003
Eye height	827.22	3297	275.74	3.74	0.012
Chest depth	1114.38	3297	371.46	14.89	0.001
Skinfold (mm)	3561.7	3297	1187.23	10.92	0.007
Sitting height	579.92	3297	193.31	3.26	0.022
Buttock–knee length	242.52	3297	80.84	3.15	0.025
Hip breadth	263.56	3297	87.85	3.35	0.019

**Table 7 healthcare-12-00109-t007:** Post hoc comparison of age groups.

**Stature Height**		**Eye Height**		**Chest Depth**		**Skin Fold**	
**Age Groups**	**(Mean) Grouping**	**Age Groups**	**(Mean) Grouping**	**Age Groups**	**(Mean) Grouping**	**Age Groups**	**(Mean) Grouping**
20–29	(160.7) A	20–29	(150.9) A	20–29	(19.2) A	20–29	(26.7) A
30–39	(158.4) B	30–39	(148) AB	30–39	(21.37) B	30–39	(32.4) B
40–49	(157.4) B	40–49	(147.8) AB	40–49	(20.4) AB	40–49	(34.78) B
50–70	(155.5) B	50–70	(146) B	50–70	(25.51) C	50–70	(26.72) A
**Sitting Height**		**Buttock–Knee Length**		**Hip Breadth**			
**Age Groups**	**(Mean) Grouping**	**Age Groups**	**(Mean) Grouping**	**Age Groups**	**(Mean) Grouping**		
20–29	(82.1) A	20–29	(53.16) AB	20–29	(35.62) A		
30–39	(79.91) AB	30–39	(54.34) A	30–39	(36.75) AB		
40–49	(78.79) B	40–49	(53.7) AC	40–49	(38.17) B		
50–70	(78.58) AB	50–70	(51.48) BC	50–70	(36.57) AB		

Age groups with the same letter are not significantly different (α = 0.05).

**Table 8 healthcare-12-00109-t008:** Anthropometric body dimensions of female adults in different Asian populations.

Dimensions	Saudi Arabian	Singaporean	Indonesian	Thai	Taiwanese	Chinese	Japanese	Korean	Iranian	Egyptian	Omani	British
weight	66.1	55	53	49.9	53.8	52	52.2	53.5	61.7	62.6	71.4	NA
stature height	158.5	161.1	162	157.9	157.3	157	156.9	158.8	158.5	160.6	157.1	160.9
eye height	148.6	150.3	150	146.3	145.7	145.4	144.8	148	147.2	149.2	147.1	149.3
shoulder height	133.1	134.3	134	129.7	128.5	127.1	127	128.9	130.9	130.6	130.9	NA
elbow height	101.2	101.8	102	99	100.7	98.7	98.3	NA	98.9	95.5	98.6	98.5
knuckle height	73.2	71.5	71	68.3	NA	NA	NA	NA	69.5	62.2	67.9	69.9
chest depth	21	22	23	20	29.3	26	28.1	NA	25.6	NA	24.4	24.9
sitting height	80.1	84.1	85	83.7	84.8	85.5	85	86.6	82.3	83.8	80.2	84.6
sitting eye height	70.4	72.7	74	73	73.5	73.9	73.2	75.8	72.3	74.3	69.6	73.8
elbow rest height	22.7	23.1	25	23.1	25.4	25.1	25.3	26.3	21.5	19.7	20.3	22.6
thigh clearance	13.2	12.3	14	12	NA	NA	NA	NA	14.8	NA	14.1	13.8
knee height	50.8	48.6	49	48.1	41.2	44.6	41.2	NA	47.6	49.9	54.3	49.7
buttock–knee length	53.5	52	54	54.5	53	52.9	53.1	52.8	56	56.5	55.1	57.1
popliteal height	45.2	42.3	40	40.2	37.9	38.2	36.2	38.4	39	NA	43.2	NA
elbow-to-elbow breadth	45.7	39.8	40	38.8	41	39.7	40.2	40.6	42.8	38.3	NA	38.6
hip breadth	36.8	30.5	35	36.2	32.2	31.7	33.3	31.9	37	36.6	38.1	37.3

**Table 9 healthcare-12-00109-t009:** Percentage of Saudi females accommodated by 900 mm supermarket receiving area height.

Criteria	Mean	Standard Deviation	Percentile for 900 mm Receiving Area Height	Conclusion
EB-150	(1004 − 150) + 25 = 879	61	63	37% much too low
EB-100	929	61	32	68% too low
EB-50	979	61	10	10% too high
EB	1029	61	2	2% much too high
				22% just right

## Data Availability

Data are contained within the article.

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
