# Peer review of "Occupational Safety and Health for Adult Saudi Arabian Women: Utilizing National Anthropometric Data"

_healthcare, 2024, doi:10.3390/healthcare12010109_

Round 1

Reviewer 1 Report

Comments and Suggestions for Authors

The article deserves to be published and it is first in kind for the population of KSA. However, this article needs further improvement. The author can improve it by incorporating the following comments:

Title: Occupational safety and health should be written instead of occupational health and safety. From engineering point of view, we look more on safety.

Abstract: the major specific findings should be mentioned in the abstract. 

Introduction:  Introduction section should mention about women occupational safety and health issues related to anthropometric data. how they are related. Should cite most recent references.

Methods: provide information on the measures taken to address potential sampling bias in your study? As this is a measurement then error analysis would validate your results. ISO15535 the standard for anthropometric database gives guidelines to calculate the percentage of relative accuracy of the collected data.

Results: Tables 1 to 4 should add the following measurements 

*Standard error of the mean (SEM) measures how well the mean of the sample in the study estimates the mean of the entire population.

*The coefficient of variation (CV) measures the variability of the mean.

*BMI (Body Mass Index), RSH (Relative Sitting Height), and BSA (Body Surface Area), need to assess which reflects the overall health status of a targeted population. 

Table 7 compares Saudi Women with other Asian women. Why only Asian need to clarify. It will be better comparing the results with few Asian and some European and some from other Arab countries like Oman, Egypt.

Comments on the Quality of English Language

Quality of the English language is fine

Author Response

Dear Reviewer,

Thank you very much for reviewing the manuscript. Please find my response in the attached file.

Best regards,

Abdalla

Reviewer 2 Report

Comments and Suggestions for Authors

We appreciate your efforts to prepare this manuscript in focus with female body of Saudi Arabia. Indeed, the manuscript has potential in team of application. However, the manuscript failed to report reliability of data. The reliability of equipment, data collection procedure and experimenter (person who measured the data) should be included in the manuscript. They are various methods and techniques (intra-inter reliability of human body dimensions) available in various research papers which are also done similar kind of work. Please kindly refer to those manuscript before you do resubmission. Without reliability test & results, the manuscript's results can't be accepted for any application. 

Comments on the Quality of English Language

Revise required 

Author Response

(The authors gave the same response as above.)

Reviewer 3 Report

Comments and Suggestions for Authors

Overall, this is an interesting paper which aims to add to current understanding regarding anthropometric dimensions of women from Saudi Arabia. This paper has some interesting findings and I commend the author for conducting this study in a country where there will have been cultural difficulties in collecting this type of data. I have provided some minor comments/suggestions for the author to consider below:

Introduction - the author suggests that anthropometric data has only been collected for the purposes of workplace design relatively recently, however, anthropometric studies have been conducted for design (clothing design, ergonomics, etc.) for decades. The author should review the introduction to make this more clear.

Methods:

- were the two female experimenters who collected the body measures trained according to any recognised accreditation bodies, e.g. ISAK?

- Why was only the triceps skinfold site collected? Was this due to difficulties in collecting skinfold measures from other sites?

- section 2.3 Equipment - I agree that in many cases 3D scanning devices are expensive, however, several lower cost devices have been developed which is increasing accessibility to this technology. Also, the author states that 'traditional measures yielded anthropometric data that was shown to be just as dependable and accurate.' Was a validation/agreement study conducted comparing measures collected using manual and 3D scan based measures to support this statement?

Results:

- Section 3.1 - could not all of this data have been presented in a single table? This would improve the readability of the paper.

- Figure 5 - the author should avoid using 3D effects when presented graphs as it does not improve the interpretation of data.

- It is unusual that the author has combined the results and discussion sections in this manuscript. Typically these sections would be presented separately, with the simply presented in the results section and then interpreted in the discussion section. This is particularly relevant to section 3.4, since this is quite an interesting discussion of the findings which shows the importance of detailed anthropometric datasets such as this. However, a large portion of the manuscript is devoted to this section, even though it was not stated as being a primary objective of the study to analyse cashier workstations. I would therefore suggest that this be reduced to a section of the discussion to highlight the potential implications/use of these measures.  

Author Response

(The authors gave the same response as above.)

Round 2

Reviewer 2 Report

Comments and Suggestions for Authors

The revised manuscript has changed with suggested comments. Henceforth, I recommended for publication